# Cell-Type-Specific Gene Regulatory Networks of Pro-Inflammatory and Pro-Resolving Lipid Mediator Biosynthesis in the Immune System

**DOI:** 10.3390/ijms24054342

**Published:** 2023-02-22

**Authors:** Matti Hoch, Jannik Rauthe, Konstantin Cesnulevicius, Myron Schultz, David Lescheid, Olaf Wolkenhauer, Valerio Chiurchiù, Shailendra Gupta

**Affiliations:** 1Department of Systems Biology and Bioinformatics, University of Rostock, 18055 Rostock, Germany; 2Heel GmbH, 76532 Baden-Baden, Germany; 3Leibniz-Institute for Food Systems Biology, Technical University of Munich, 85354 Freising, Germany; 4Stellenbosch Institute of Advanced Study, Wallenberg Research Centre, Stellenbosch University, Stellenbosch 7602, South Africa; 5Institute of Translational Pharmacology, National Research Council, 00133 Rome, Italy; 6Laboratory of Resolution of Neuroinflammation, IRCCS Santa Lucia Foundation, 00179 Rome, Italy

**Keywords:** inflammation resolution, network modeling, lipid mediators, RNA-seq, machine learning

## Abstract

Lipid mediators are important regulators in inflammatory responses, and their biosynthetic pathways are targeted by commonly used anti-inflammatory drugs. Switching from pro-inflammatory lipid mediators (PIMs) to specialized pro-resolving (SPMs) is a critical step toward acute inflammation resolution and preventing chronic inflammation. Although the biosynthetic pathways and enzymes for PIMs and SPMs have now been largely identified, the actual transcriptional profiles underlying the immune cell type-specific transcriptional profiles of these mediators are still unknown. Using the Atlas of Inflammation Resolution, we created a large network of gene regulatory interactions linked to the biosynthesis of SPMs and PIMs. By mapping single-cell sequencing data, we identified cell type-specific gene regulatory networks of the lipid mediator biosynthesis. Using machine learning approaches combined with network features, we identified cell clusters of similar transcriptional regulation and demonstrated how specific immune cell activation affects PIM and SPM profiles. We found substantial differences in regulatory networks in related cells, accounting for network-based preprocessing in functional single-cell analyses. Our results not only provide further insight into the gene regulation of lipid mediators in the immune response but also shed light on the contribution of selected cell types in their biosynthesis.

## 1. Introduction

Inflammation is a complex and tightly regulated process that protects the body from any form of damage, insult, or infection [1,2,3]. In addition to secreted proteins (cytokines), lipid mediators (LMs) generated from polyunsaturated fatty acids (PUFAs) in the cell membrane play a key role in regulating all the phases of inflammation, from the initial acute response to its fine-tuning of inflammation transition and even termination [4]. During acute inflammation, arachidonic acid (AA) is the main PUFA that is used for the biosynthesis of over 150 different pro-inflammatory lipid mediators (PIMs) (i.e., various classes of prostaglandins, leukotrienes, and thromboxanes) that altogether act as the “fire-starters” of the inflammatory response by controlling vascular and cellular responses and by determining the cardinal signs of inflammation (redness, heat, swelling, pain, and loss of function) [5,6,7,8,9]. In the last two decades, various LMs involved in the termination of inflammation, so-called “specialized pro-resolving mediators (SPMs), have been identified and are composed of over 30 lipids derived from ω-3 PUFA such as docosahexaenoic acid (DHA) and eicosapentaenoic acid (EPA) [10,11,12,13]. Unlike PIMs, SPMs promote the resolution of inflammation and tissue repair by activating the cardinal signs of resolution (removal, restoration, regeneration, remission, and relief). Their tightly regulated synthesis during the inflammatory response is a crucial step in extinguishing the fire of inflammation, thus favoring the return to homeostasis, as well as in the prevention of excessive inflammatory responses and the development of chronic inflammation [9,13,14,15].

Although acute inflammation involves a large number of cells and molecules, its initiation triggers relatively straightforward and ubiquitous cascades of various strengths depending on the type and amount of stimulus (e.g., production of PIMSs, vasodilation, chemotaxis of various immune cells) that ensure a rapid response in any tissue [1,3]. In contrast, the resolution of inflammation mechanisms (e.g., type and levels of SPM production and their downstream signaling cascades) strongly depends on the tissue microenvironment [9,12]. Although the SPM biosynthetic pathways, including regulatory enzymes, are now largely identified and are the very same as those involved in PIM production, the actual regulatory processes underlying cell-type-specific mediator profiles remain elusive. In 2018, Norris and Serhan performed a metabolipidomics analysis of human whole blood and identified functional and cell type-specific LM profiles [16]. Their results showed that haematopoietically and functionally distant cell types have similar LM profiles and, vice versa, closely related cells can synthesize substantially different LMs, indicating individual cell type-specific regulations. LMs are secreted to neighboring cells in an auto- and paracrine fashion [17,18]. Such a highly localized response would require cell-type-specific transcriptional programs and thus a cell-type-specific expression of transcriptional regulatory networks.

Usually, cell types are defined by cell-type-specific markers, morphological features, and functional properties or by their distinct (multi-)omics profiles [19]. With the advancement of single-cell RNA sequencing (scRNA-Seq), new subsets of existing cell types are constantly being defined, and the established boundaries between cell types seem to disappear [20]. Thus, modern experiments focus on single-cell data rather than bulk samples of apparently related cells. However, the idea of subsets of a defined cell type also adds new complexity to understanding cell-type-specific signal transduction that distinguishes them from others. To address the challenge of analyzing physiological or functional relationships in single cells, unsupervised machine-learning approaches proved to be extremely useful for identifying patterns in single-cell expression profiles [21,22]. In addition to clustering cells based on their omics profiles, generating topological features from cell-type-specific molecular interaction networks enable the study of functional relationships between molecules and genes [23]. However, the analysis capabilities rely on the causal interactions in the network, making network construction and curation an essential step.

Recently, we published the Atlas of Inflammation Resolution (AIR) as a publicly available, web-based knowledge platform of molecular interactions and biological processes involved in acute inflammation and its resolution [24]. We have identified key processes at each stage of inflammation and developed a standardized representation of the associated molecular interactions in so-called standardized molecular interaction maps (MIMs). The manually curated causal interactions enable the use of systems biology approaches to infer regulatory circuits, predict signal transduction pathways, or perform perturbation experiments [25]. Among others, the AIR provides a detailed description of the biosynthetic pathways of PIMs and SPMs from their precursors AA, DHA, and EPA. In this study, we investigated cell-type-specific transcriptional networks associated with LM synthesis pathways. We mapped scRNA-Seq data to gene regulatory networks extracted from the AIR and examined how the networks are affected by differences in the expression of transcription factors. We investigated how cellular LM profiles are modulated by changes in the cell-type-specific network topology of gene regulatory networks (GRNs). By applying unsupervised machine learning approaches to network topological features extracted from the GRNs, we clustered single cells according to their regulatory mechanisms and identified their key gene regulators. We have shown how the application of network-based approaches can improve the analysis of functional molecular pathways and their regulatory networks using scRNA-Seq data. Our results shed light on the gene regulation of LM synthesizing enzymes across various immune cell types.

## 2. Results

### 2.1. Cell-Type-Specific LM Pathways

We clustered the cells based on the expression profile of genes included in the AIR database, i.e., being directly related to immunological processes (Figure 1A and Figure 2A). The dimensionality reduction largely restored the cell type clusters as they are defined in the metadata of both datasets. We investigated the expression of LM enzymes in the cells, and whether clusters of enzyme expression correspond to Uniform Manifold Approximation and Projection (UMAP) clustering. Additionally, for each cell, we analyzed whether the substrates of LM biosynthesis, AA, DHA, or EPA, are linked to the final products through the expression of catalyzing enzymes.

The GSE122108 dataset consists of mononuclear phagocytes, mainly macrophages, of different tissues, with various pro- and anti-inflammatory stimuli. The cell types with fewer samples, such as monocytes, dendritic cells, and microglia cells, were partially restored (Figure 1A). Macrophage samples are widely scattered and partially mixed with the clusters of the other cell types because they originate from a wide variety of tissues. One macrophage cluster separates from all other cells and consists mainly of peritoneal cells. These peritoneal macrophages also show a distinct LM enzyme profile, with an expression of many genes and the only cells with consistently high expression of *Alox15* and *Ptgis* and, thus, are the only cell types expressing the required enzymes for all LM classes (Figure 1C). From the analysis, it emerged that while almost all cell types are fully capable to synthesize prostaglandins, leukotrienes, and thromboxanes, very few cell types can only synthesize SPMs. Indeed, lipoxins (that are generated by AA but still belong to the super-family of SPMs), protectins, and D-resolvins are produced only by macrophages, maresins only by macrophages and microglia, while E-resolvins are produced by all immune cells, including dendritic cells and monocytes (Figure 1B). Interestingly, lipoxins, protectins, and D-resolvins show a similar pattern due to the expression of the enzyme Alox15. In contrast, a group of dendritic cells expresses only those enzymes required for synthesizing E-resolvins and leukotrienes. Microglia also show a consistent expression profile, particularly of *Alox5*, *Cbr1*, *Gpx4*, and *Ptgs1*.

The GSE109125 dataset consists of many different cell types spanning the hematopoietic lineage and includes stem cells, epithelial cells, and both compartments of innate and adaptive immune cell populations, with monocytes being the only missing cell subsets. The UMAP of immune-filtered gene expression was able to restore the cell type groups to a high degree (Figure 2A). The two-dimensional projection of the UMAP graph shows the cell branching in two directions starting from the hematopoietic cell group. Except for B cells, which are placed closer to the myeloid cells, these two groups coincide with the lymphoid and myeloid lineages, respectively. The analysis revealed that the overall ability to synthesize LMs, based on the expression of required enzymes, is much lower in lymphoid than in myeloid cells (Figure 2B). In particular, cells belonging to the myeloid lineage and hematopoietic stem cells are the ones most capable to biosynthesize both PIMs and SPMs, with macrophages and granulocytes (neutrophils, basophils, and eosinophils) being the most efficient due to the high expression of LM enzymes (Figure 2C). Mast cells and ILCs show a similar biosynthetic pathway in producing PIMs and only one class of SPMs, i.e., E-resolvins. As expected, NK cells and NKT cells also share a similar ability to synthesize the same class of LMs, which are limited only to prostaglandins (except for I-prostaglandins) and thromboxanes; however, only NKT cells can produce maresins. Interestingly, epithelial cells display a biosynthetic pathway identical to NKT cells. Of note, it seems that neither T cells nor B cells are capable produce any LMs.

### 2.2. Cell-Type-Specific Gene Regulation

Despite apparently similar expression profiles of LM enzymes, cells may differ in transcriptional circuits that tightly regulate LM synthesis. Moreover, a similar expression profile may be regulated by substantially different transcription factor networks, which would be required for cell-type-specific responses to stimuli in different tissues. Thus, we analyzed the connectivity between transcription factors and enzymes of each LM class in the cell-type-specific GRNs. After dimensionality reduction for all classes, the embeddings were combined and projected into single UMAPs for each dataset (Figure 3A,B). For each cluster, we identified the genes with the most significant differences compared with all other cells (adj. *p*-value < 0.05, see methods). Detailed information on all clusters, their predicted genes, and included samples are available in the Appendix A.

In the GSE122108 dataset, we observed many separate clusters and good restoration of the main cell types, i.e., dendritic cells, macrophages, microglia, and monocytes (Figure 3A, Appendix A). Of note, macrophages appeared as smaller clusters that were partially composed of tissue-specific cells, e.g., from the aorta, heart, or liver. We identified the significant (adj. *p*-value < 0.05 for any LM class) genes of the microglia cells, which build the most defined cluster in the UMAP plot (Figure 3C). For the two highest-ranked genes, *Mef2a* and *Xrcc5*, we additionally showed their regulatory score in relation to their expression in all samples. The plots show how the score is significantly increased in the microglia cells and, especially for *Mef2a*, is independent of its expression. In the literature, information on tissue-specific transcriptional regulation of LM biosynthesis is very sparse. Hence, to compare our results with experimental data, we searched the literature for any evidence supporting the immune modulatory function of the genes related to microglia. Of the thirteen genes, we found clear evidence in the literature for eight genes on their relevance in microglial function and neuronal inflammation (*Mef2a* [26], *Hdac11* [27,28], *Smad3* [29], *Mef2c* [30], *Arid1a* [31,32], *Zfhx3* [33,34], *Ets1* [35], and *Jun* [36]). Four genes were mentioned in experiments on microglial inflammation (*Xrcc5* [37,38], *Zfp191* [39], *Prdm1* [40], and *Usf2* [41]), whereas no information was found in the literature for only two genes (*Znf383* and *Nfrkb*). The mode of action of the predicted genes in modulating microglia function has been attributed to their influence on cytokine expression. Our results suggest that they modulate the immune response by also regulating the expression of enzymes involved in the biosynthesis of LMs. *Smad3*, *Jun*, *Usf2*, and *Xrcc5* have already been described in their regulation of prostaglandins, while little to no research is available on the other LM classes [42,43,44,45]. *Mef2a* and *Mef2c* have been identified as downstream effectors of PGE2, which could indicate a feedback loop on prostaglandin e synthesis [46,47].

In contrast, in the GSE109125 dataset, the original cell types are more heterogeneously distributed between clusters (Figure 3B, Appendix A). The differences in the expression of immune-related genes between the major immune cell types are not reflected in the TFs associated with the LMs. However, two clusters consisting of hematopoietic stem cells and mast cells, respectively, are strongly separated. While no significant TFs were identified for the latter, the former shows a division into three subclusters, from each of which several significant TFs were identified. Interestingly, based on cell metadata, the three subclusters appear to represent stages of lymphoid hematopoiesis, namely (i) bone marrow-derived stem cells (BMSCs) followed by (ii) early (DN1 and DN2a lymphocytes) and (iii) late lymphoid progenitor cells. While BMSCs express many LM enzymes, they are downregulated in lymphoid progenitors. When comparing the regulatory scores of stem cells and early lymphoid progenitor cells, *Hlf* had the greatest difference in its score for all LM classes (not shown). *Hlf* is an important regulator of lymphoid development in the hematopoietic lineage [48]. Our results suggest that modulation of LM synthesis by gene regulation of LM enzymes may play a role in shaping the fate of lymphoid cells by *Hlf*.

### 2.3. Immune Cell Activation Modulates Gene Regulatory Networks of Lipid Mediators

Several samples in the GSE122108 data were treated with pro- or anti-inflammatory stimuli at several time points, including lipopolysaccharide stimulation (LPS), *C. albicans* infection, induction of injury, paracetamol, and thioglycolate. We compared the cells at successive time points for each stimulus and identified the TFs with the strongest changes in their gene regulatory activity for each LM class (Figure 4A). For the selected genes, we additionally show violin plots comparing their expression values (read counts) and topology scores, showing that the estimated change in connectivity is independent of their expression (Figure 4B). In general, the predicted that TFs show a strong variability between cells and the different stimuli, suggesting that gene regulation of LMs in the immune response is highly cell-type and environment specific. Additionally, especially at early time points, the identified TFs also differ substantially between PIMs (e.g., the prostaglandin classes) and SPMs (e.g., the resolvin classes) due to the distinct enzyme profile, arguing for fine-tuned gene regulation. At later time points, the difference between PIM and SPM classes becomes smaller, and the number of overlapping TFs increases.

Many predicted genes are well-known regulators of the immune response to respective stimuli. For example, in liver macrophages stimulated with APAP, Hes1 appears to be a key regulatory TF of most SPM classes. In vivo experiments showed that blocking the Notch signaling pathway in mice reduced *Hes1* levels and increased susceptibility to APAP-induced liver injury [49]. In thioglycolate-stimulated monocytes/ macrophages, our model predicted several genes related to both PIMs and SPMs synthesis, which have also been described in the literature, such as *Epas1* (prostaglandins), *Egr2* (prostaglandins), *Cebpb* (all LM classes), and *Srebp1* (SPMs). *Epas1*, coding for HIF-2α, is an important mediator of cellular processes and macrophage recruitment in response to hypoxia [50]. In an experimental thioglycolate periodontitis model, *Egr2* and *Cebpb* were required for macrophage activation [51]. In *Srebp1* knockdown mice, thioglycolate-elicited macrophages showed increased levels of pro-inflammatory cytokines and reduced levels of DHA and EPA during the resolution phase after *Tlr4* activation [52]. Although being related cell types, the five subtypes of LPS-stimulated lung macrophages also differ in the predicted TFs. Two subtypes of lung macrophages originate from broncho-alveolar lavage (BAL) and show a similar gene regulation of prostaglandins through *Klf10* and *Vhl*. Both genes have already been associated with inflammatory responses in BAL macrophages [53,54]. For the other LMs, both BAL subtypes do not overlap in the predicted TFs. The remaining lung macrophage subtypes are defined by cell sorting markers. Their samples for which data are available on days zero and three after LPS stimulation overlap at *Stat1*, *Stat2*, and *Pias1*. The results become more diverse at later time points (day six vs. day three). We observed that the three MHC-II^-^ macrophage and monocyte subtypes partially overlap in *Foxk2*, *Rora*, and *Ing4* genes that are associated with cytokine production in response to LPS [55,56], while for the MHC-II^+^ subtype, we predicted autophagy-related genes *Rb1cc1*, *Rb1*, and *Hdac2* [57,58,59]. Whether or not this difference is caused by MHC-II is yet to be determined, as only limited evidence connects MHC-II with the predicted genes.

### 2.4. LM Gene Regulation Shows Substantial Differences in Related Cell Types

Since the transcriptional regulation of LMs appears to be tightly regulated and cell-type-specific, we investigated the extent to which closely related cell types may differ in the transcriptional interaction networks of PIM and SPM synthesis. We identified the cell pairs with the smallest distance in expression-based UMAP but the largest distance in transcriptional network-based UMAP. The top-ranked sample pair consists of a macrophage from the aorta and a macrophage from the lung stimulated with LPS (Figure 5A). Both tissue-specific subtypes of macrophages appear to have a nearly identical transcriptomic profile but substantially differ in LM gene regulation. Thus, we extracted the core regulatory networks (CRNs) to gain further insight into the genes contributing to the observed differences (Figure 5B) and we additionally generated a CRN of an unstimulated sample of the same lung macrophage subtype but without LPS stimulation to ensure that the difference is not caused by the response to LPS. Interestingly, the CRN shows that the expression of most LM enzymes is similar except for *Ptgs2*, which is not expressed in aorta macrophages. In contrast, *Ptgs2* is highly expressed in aorta macrophages with high expression levels of the TFs *Jun*, *Egr1*, and *Fos*. All these three genes are highly associated with atherosclerotic inflammation [60,61,62]. *Egr1* is involved in the response to mechanical or oxidative stress and, thus, the development of atherosclerosis from plaques and hypertonia [60,63,64].

## 3. Discussion

The immune response is a tightly regulated system involving a large number of different cell types with specific spatiotemporal functions. Over the years, experimental research has attempted to identify and describe the molecular and functional processes involved. However, although more and more knowledge is being gained and regulatory processes are being elucidated, increasing complexity is blurring the boundaries between cell types. At the same time, it is challenging to study the role of specific cells in immunological processes and cell-type-specific immune responses. One reason for this is the enormous cost and effort required to study the effects of a single transcription factor, e.g., using gene knockout or targeted inhibition of transcripts with miRNAs. Consequently, experimental identification of novel transcription factors regulating a particular process is not feasible and, therefore, tends to be targeted based on hypotheses from other experiments. Moreover, experimental data are mostly generated by measurable changes, such as changes in their expression using RNA-Seq, but TFs do not necessarily have altered expression themselves, and cell-type-specific changes could be mediated by changes in the topology of gene regulatory networks. As a result, very little information on cell-type-specific gene regulation can be found in the literature, especially for the relatively young field of LM biosynthesis.

While the effects of LMs in cells and tissues have been extensively studied, particularly for PIMs but recently also for SPMs, the regulatory mechanisms underlying their biosynthesis in a cell-type-specific manner is still not very well investigated, which may be important to understand how various cells communicate to resolve inflammation. This complexity of the LM response is also shown by the ability of myeloid cells (i.e., macrophages and granulocytes) to synthesize both PIMs and SPMs, while lymphoid cells seem incapable to produce any LM. These results are also supported by the vast literature where both classes of pro-inflammatory and pro-resolving LMs have been detected in a low or high picomolar range in most cell populations belonging to the myeloid and innate compartment of immunity. In contrast, evidence that cells of the lymphoid and adaptive immune system can produce such LMs is very scarce (extensively reviewed in [8,9,65,66]).

Here, we investigated LM synthesis at the transcriptional level using in silico analyses of cell-type-specific gene regulatory networks from scRNA-Seq data. Our results highlight that, although cell types have similar expression profiles, they might exhibit distinct transcriptional regulations of LM synthesis and, thus, respond with different LM productions to experimental conditions. For instance, the higher expression of the stress- and inflammation-related genes *Egr1*, *Jun*, and *Fos* in aorta macrophages than in lung macrophages and their association with LM gene regulation, despite their similar RNA-Seq profiles, might account for a physiological advantage in the aorta by enabling a sufficient LM response to stress stimuli, such as hypertonia. Thus, our study showed that systems biology approaches could identify cell- and tissue-specific patterns of gene expression–phenotype relationships. Correlating the measured gene expression with underlying gene regulation can improve the analysis and interpretation of scRNA-Seq data.

While large numbers of gene regulatory interactions are available in public databases, identified using in silico predictions of binding motifs, information on the type and strength of these interactions is rather scarce. Even if available, including such information also introduces new challenges, such as integrating competitive TF interactions. As our study aims to compare cell-type-specific GRNs, we built the networks using qualitative data (considering whether there is an interaction between a TF and a gene) to avoid false negative information and, consequently, disruptions in the network. By integrating expression data and topology algorithms, the qualitative information is converted into quantitative regulation scores for machine learning algorithms, providing a valuable estimation of a TF’s relevance in the GRN. Similar in silico studies on gene interaction networks showed the use of network topology information to predict key regulators and motifs [67,68,69]. The resulting bias towards highly connected nodes was encountered by normalizing the regulation score by the node degree. In our approach, we include information on multiple genes per LM class in the calculation of regulatory scores as well as combining the predictions from machine learning for multiple LM classes. The approach can be translated equally to other immune mediators, such as cytokines. The interpretability of the molecular results of this study is further limited to mice, although the methodology can be easily translated into human data. We specifically chose murine RNA-Seq data as much more murine than human in vivo studies are available that provide experimental evidence on gene-to-phenotype associations.

With our study, we provided examples of how network-based scRNA-Seq data analyses could provide insights into cellular mechanisms of LM regulation and generate new hypotheses for follow-up investigations using human data. Thus, our results account for integrating systems biology approaches to stratify cellular responses more accurately in experimental settings and to discriminate or predict pathological states based on the ability of specific disease-associated cells to engage in pro-inflammatory or pro-resolving pathways.

## 4. Materials and Methods

### 4.1. Network Curation

We extracted molecular interactions from the “lipid mediator biosynthesis from arachidonic acid” (Appendix A), “lipid mediator biosynthesis from DHA” (Appendix A), and “lipid mediator biosynthesis from EPA” (Appendix A) submaps of the AIR using its Xplore tool. The maps were then extended with transcription factor (TF) and gene target interactions from the AIR MIM to create a gene regulatory network (GRN). Catalytic reactions were transformed into the activity flow format by integrating enzymes in between the source and target element with positive interactions each (Figure 6A). The resulting network can be considered as the graph G of a set of elements (vertices V(G)) and connecting interactions (edges E(G)). The edges encode whether two elements are linked by (de)activation, up-, or downregulation and are defined as a collection of triples E⊂(s×r×t) consisting of a source element s∈V, a relation r∈{−1,1}, and a target element t∈V.

### 4.2. Data Processing and Integration

Two murine single-cell RNA-seq profiles (GSE122108 and GSE109125) with preprocessed and library-size normalized read counts (*q*) by the Immunological Genome (ImmGen) Project were downloaded from their website (http://rstats.immgen.org/DataPage/, accessed on 10 November 2022). They include many different immune cell types from various tissues with extensive descriptions of the samples’ origins and sorting markers. Both datasets have been described in detail in their respective published studies [70,71]. While the GSE122108 dataset consists only of phagocytotic mononuclear cells, mainly macrophages and monocytes, the GSE109125 data includes cells from all major cell types of the lymphoid and myeloid lineage. We mapped the murine genes from the data with genes in the AIR using human–mouse gene identifier associations from the Ensemble database (https://www.ensembl.org/, accessed on 23 August 2020). We defined a read count of 10 as a threshold to mark a gene as expressed or unexpressed which is slightly higher than the threshold of 5 used by the ImmGen project to exclude more genes with non-functional expression levels [71,72]. Genes with read count values below the threshold in a cell type *c* were removed from G resulting in cell-type-specific subgraphs $G_{c}$ with $V_{c}\subseteq V$ and $E_{c}\subseteq E$ (Figure 6B). Proteins from the manually curated submaps, i.e., enzymes directly involved in the LM biosynthesis, as well as elements with no expression, such as metabolites or phenotypes, were not removed from *G_c_*. For cellular normalization, we divided the read count value of each gene by its highest absolute value across all cell types, resulting in the cell type normalized read count $\hat{q}$.

### 4.3. Topological Analysis

A path P in the MIM of the length l∈ℕ can be written as the sequence $\left(u_{1}\overset{r_{1}}{\to }u_{2}\overset{r_{2}}{\to }\ldots \overset{r_{L}}{\to }u_{l+1}\right)$ with $\left(u_{i},r_{i},u_{i+1}\right)\in E$. The relation 
r∈{−1,1}
between the first and final element of any P is defined as $\left(r_{1}\cdot r_{2}\cdot \ldots \;\cdot r_{l}\right)$ for all interactions along P. The shortest path SP between (*u*, *v*) is defined as an existing path $P_{u,v}$ between u and v where $l\left(P_{u,v}\right)$ is minimized. In each subgraph $G_{c}$, the shortest paths between precursors and the final products in the LM biosynthesis were identified using the Breadth-First-Search. In addition, pathing algorithms were applied to identify core regulatory networks (CRNs), which are combined pathways from genes to LM enzymes with the maximum score of genes passed. The identification of CRNs becomes a widest path problem and was solved with an adaptation of Dijkstra’s algorithm. The edge weights are based on the edge’s target node u and were set to either ${\bar{s}}_{u}$ for CRNs of a single cell or $\left|\Delta {\bar{s}}_{u}\right|$ when comparing two sets of cells.

### 4.4. Topological Weighting

For each LM class p, we calculated a weighting factor for all elements in the submaps representing their topological inclusion in the paths connected to p. We recently described this weighting approach [25]. In summary, the weighting of an element e is calculated based on the percentage of elements and paths connected to p. $N_{paths}$ is the number of all paths to *p* and $N_{paths_{e}}\subset N_{paths}$ are paths that go through e. $N_{nodes}$ is the number of elements connected to p and $N_{nodes_{e}}\subset N_{nodes}$ the number of elements on the path from e to p:$$w_{e,p}=r\left(SP_{e,p}\right)\cdot \left(\frac{N_{paths}{}_{e}}{N_{paths}}+\frac{N_{nodes}{}_{e}}{N_{nodes}}\right)$$

### 4.5. Feature Extraction

We generated a regulatory score
$\bar{s}$
for each gene in $G_{c}$, representing its association to LM synthesis. We performed a stepwise signal propagation based on the approach presented by Lee and Cho [73], starting from the LM enzymes and continuing in the reverse direction through the transcription network (Figure 7A). The transcription factors’ scores were updated at each step based on degree centralities (=number of interactions) in the original GRN, their targets’ scores in the previous step, and their normalized read count $\hat{q}$ (Figure 7B). The simulation was performed for each cell type and initiated separately for each LM class by setting the starting scores $s_{e}^{t=0}=w_{e,p}$ for each enzyme e in the LM class p. The final regulatory score for each node u in the network is then defined as the area under the curve (AUC) of scores over 100 signaling steps: ${\bar{s}}_{u}=\int_{t=0}^{100}s_{u}^{t}$ (Figure 7C).

### 4.6. Cell Type Clustering

We performed a Uniform Manifold Approximation and Projection (UMAP) analysis for both datasets using both the filtered expression data and, for each LM class, using the regulatory scores $\bar{s}$ generated from $G_{c}$ (Figure 6C). UMAP reduces the high dimensionality of the input data into a two-dimensional graphical representation where each point corresponds to a cell in the data. In this way, cells with similar values are positioned close to each other, while separated cells indicate larger differences. Cell clusters were identified using manually adjusted k-means clustering on the generated embeddings. To visualize distributions across all LM classes, their embeddings were combined into a single dataset and a new UMAP was performed. Clustering in the enzyme expression heatmaps was performed using the Euclid-based hierarchical clustering method of the Python package seaborn version 0.12.1 [74].

### 4.7. Statistical Evaluation of Features

The goal of the statistical analysis is to identify features that differ in a group of samples, i.e., clusters. Since the calculation of regulatory scores is based on the expression of the feature in the cell, the final scores are biased towards $\hat{q}$. Therefore, instead of calculating the highest scores, the features should be analyzed in relation to $\hat{q}$. In an LM class, the $\hat{q}$ and $\bar{s}$ values of a gene in all cells, which are not in the cluster, were fitted to linear regression, and a half-normal distribution was created from the absolute distances of each cell from the line (Figure 7D). The *p*-value of the feature in the cluster is then calculated from the z-score of the average distance of the cluster’s cells in the distribution.

## 5. Conclusions

In conclusion, this study demonstrates how the application of network-based approaches enables the identification of cell-type-specific regulatory networks from scRNA-Seq data. We showed that gene regulation of the lipid mediator biosynthesis is highly dependent on the cell type and stimuli. Our results further argue for a fine-tuned transcriptional modulation of immune cell types and emphasize the necessity of systems biology approaches in understanding the underlying mechanisms.

## Figures and Tables

**Figure 1 ijms-24-04342-f001:**
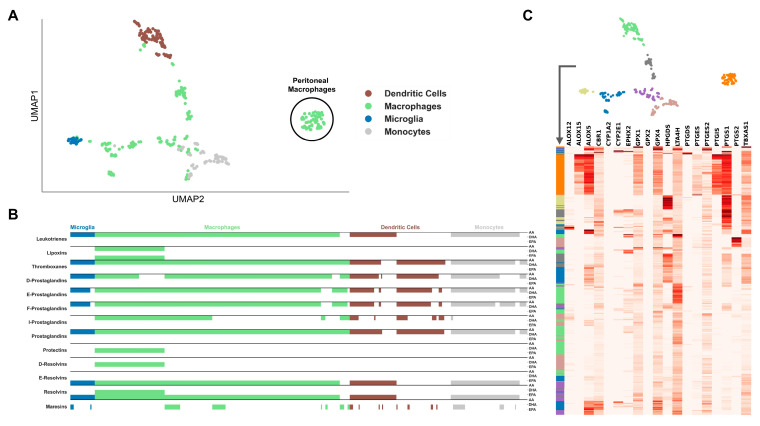
Clustering of immune cell types in the GSE122108 dataset. (**A**) UMAP plot of immune cell scRNA-seq data with highlighted clusters based on scRNA-seq cell sorting. Genes in the dataset were filtered for those included in the “Atlas of Inflammation Resolution” database. (**B**) Cell type-specific de novo biosynthetic pathways of each lipid mediator class from the precursor molecules arachidonic acid (AA), docosahexaenoic acid (DHA), or eicosapentaenoic acid (EPA) based on the expression of catalyzing enzymes. (**C**) Clustered heatmap of lipid mediator enzyme expression color-coded by clusters defined from the UMAP in (**A**).

**Figure 2 ijms-24-04342-f002:**
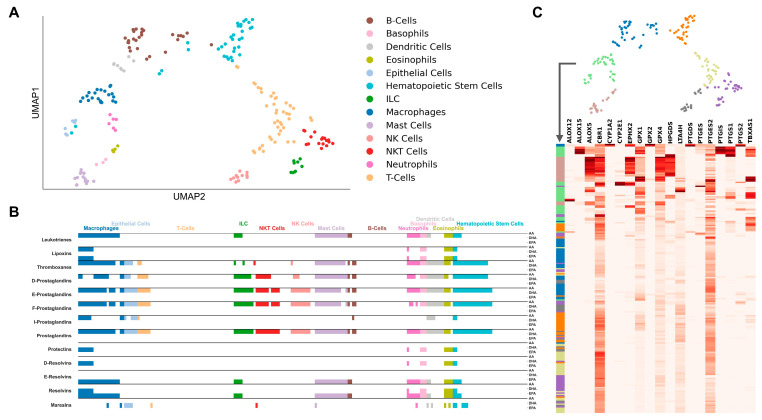
Clustering of immune cell types in the GSE109125 dataset. (**A**) UMAP plot of immune cell scRNA-seq data with highlighted clusters based on scRNA-seq cell sorting. Genes in the dataset were filtered for those included in the “Atlas of Inflammation Resolution” database. (**B**) Cell type-specific de novo biosynthetic pathways of each lipid mediator class from the precursor molecules arachidonic acid (AA), docosahexaenoic acid (DHA), or eicosapentaenoic acid (EPA) based on the expression of catalyzing enzymes. (**C**) Clustered heatmap of lipid mediator enzyme expression color-coded by clusters defined from the UMAP in (**A**).

**Figure 3 ijms-24-04342-f003:**
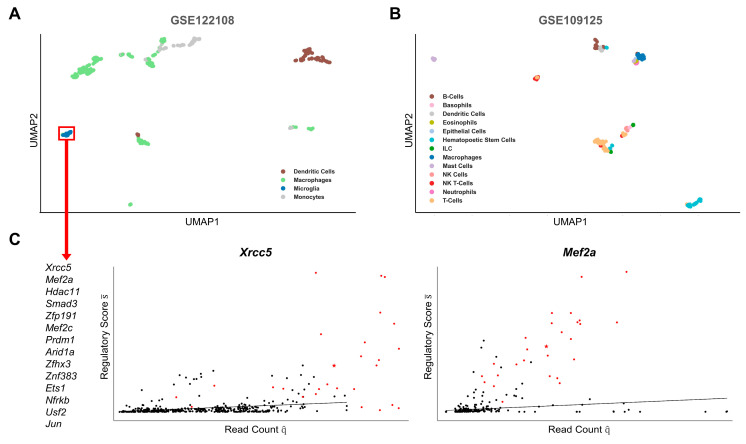
UMAP clustering of individual cells based on their topological association and expression of transcription factors related to lipid mediator biosynthesis. scRNA-Seq profiles of two data sets, GSE122108 (**A**) and GSE109125 (**B**) were mapped to a gene regulatory network and topological features were extracted for the UMAP. (**C**) For the microglial cell cluster, transcription factors with significantly higher scores than other clusters are shown. For the two highest-scoring transcription factors, *Xrcc5* and *Mef2a*, their score, and their expression in the cluster (red) compared with all other cells (black) are shown in a scatter plot.

**Figure 4 ijms-24-04342-f004:**
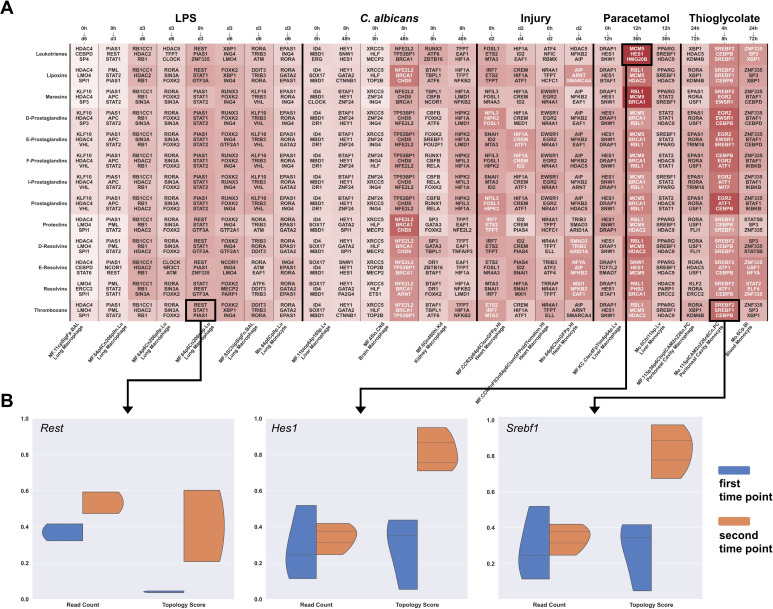
Transcription factors associated with stimulation of immune cell types. (**A**) The GSE122108 dataset includes gene expression data of immune cell types stimulated with inflammatory agents for different time points. We identified the three major transcription factors with increasing topological association to each lipid mediator class between time points of each cell type. (**B**) For three selected genes, *Rest*, *Hes1*, and *Srebf1*, we show the normalized expression levels and topology scores for all samples in a violin plot.

**Figure 5 ijms-24-04342-f005:**
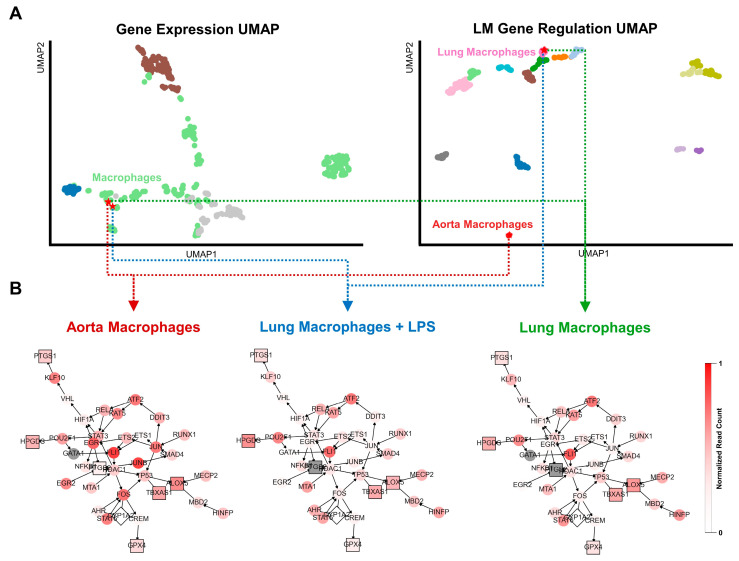
Identification of cells with similar expression profiles but different transcriptional regulation of lipid mediators (LMs). (**A**) Cell samples with minimized distance within the expression-based and large distance in the LM-regulation-based UMAP. The highest-ranked cell pair consists of a sample from aortic macrophages and one from lung macrophages stimulated with LPS. We added an unstimulated lung macrophage to show that the difference is not caused by the reaction to LPS. (**B**) Gene regulatory networks connected to LM enzymes for all cells colored by their normalized read count values. The shape of the nodes distinguishes between TFs (round), LM enzymes (square), and nodes not included in the transcriptomics data (diamond). Nodes with read counts below the absolute threshold of 10 are highlighted in gray to distinguish them from lowly expressed ones.

**Figure 6 ijms-24-04342-f006:**
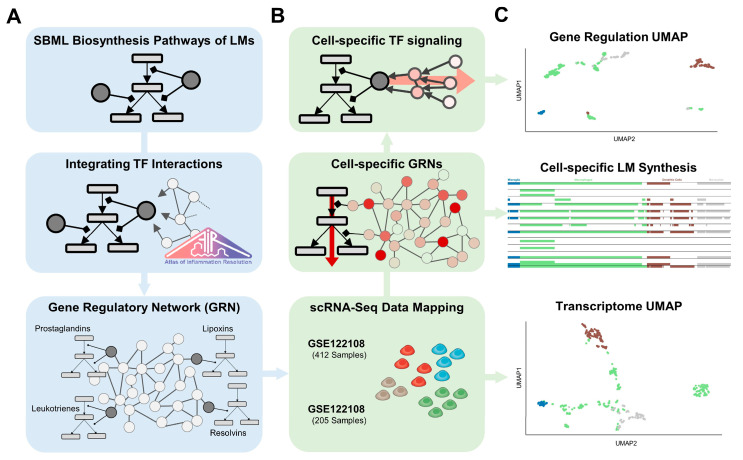
Workflow to create and analyze cell-type-specific networks on lipid mediator (LM) biosynthesis from scRNA-seq data. (**A**) Biosynthesis Pathways of LMs from AA, DHA, and EPA were merged with transcription factor gene interactions from TRRUST to create a large-scale gene regulatory network (GRN) (**B**) Cell-type-specific GRNs were then generated by mapping scRNA-Seq to genes and removing under-expressed genes from the network. From the GRNs, regulatory scores were generated for all transcription factors based on their connectivity to lipid mediator enzymes. (**C**) From the cell-type-specific GRN, we analyzed LM synthesis paths based on the expression of enzymes. Dimensionality reduction of regulatory scores for all TFs using an unsupervised machine learning approach then identified cell clusters with similar gene regulatory networks. A comparison of the resulting clusters with the clusters generated using the gene expression data allowed the identification of functionally and hematopoietically related cells.

**Figure 7 ijms-24-04342-f007:**
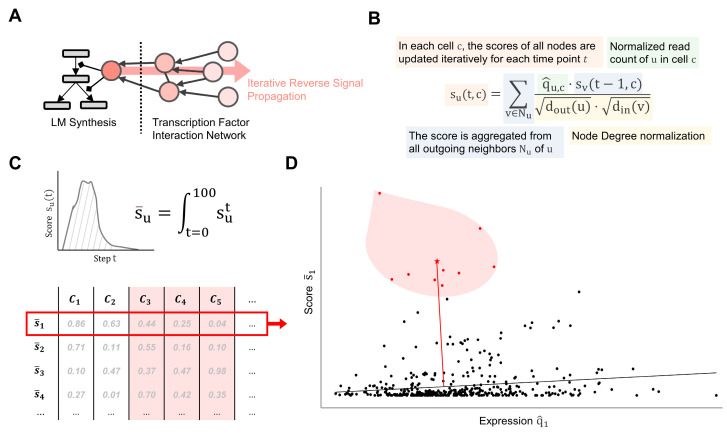
Feature extraction from the cell-type-specific gene regulatory networks (GRNs). (**A**) Starting from lipid mediator enzymes, the starting signal is traversed in reverse throughout the GRN. (**B**) For a distinct number of steps, a score is updated for each transcription factor (TF) based on its gene expression, the score of its target in the previous step, and the node degrees of both the TF and its target. (**C**) The final score is defined as the AUC of the scores throughout 100 steps and is used as an input for the unsupervised machine learning algorithm. (**D**) The statistical significance of a score is calculated based on its distance to a regression line representing the correlation between the score and the expression of the TF across cells.

## Data Availability

The two transcriptomics datasets analyzed in this study are available on the Gene Expression Omnibus (accession IDs GSE122108 and GSE109125) as well as the ImmGen databrowser (http://rstats.immgen.org/DataPage, accessed on 10 November 2022). The molecular interaction map was extracted from the “Atlas of Inflammation Resolution” (https://air.bio.informatik.uni-rostock.de, accessed on 27 December 2022). Python scripts and formatted data files are available on GitHub (https://github.com/sbi-rostock/Hoch_et_al_LM_GRNs, accessed on 27 November 2022).

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
