# Peer review of "Cell-Type-Specific Gene Regulatory Networks of Pro-Inflammatory and Pro-Resolving Lipid Mediator Biosynthesis in the Immune System"

_ijms, 2023, doi:10.3390/ijms24054342_

Round 1

Reviewer 1 Report

The authors present an in silico study aimed at studying the complex molecular mechanisms that regulate the transcription of lipid mediators with anti-inflammatory activity in different cell types of the immune system. The study is interesting and highlights the importance of deep learning and machine learning in the study of biological phenomena.

Comments:

1) A flow-chart would be helpful to better understand the methodological process followed by the authors.

2) The data analyzed are from murine datasets. The authors should comment on the possibility of conducting such studies in humans.

3) It is unclear why monocytes were not included in the analysis

4) Comment on the diagnostic and therapeutic utility of in silico studies and whether they require further in vivo experimental verification is helpful.

5) It would be interesting to know whether such studies are applicable not only to lipids but to mediators such as cytokines and for whose blockade we have effective tools for treating chronic inflammation

6) Several errors regarding missing bibliographic references are pointed out in my review copy

Author Response

We thank the reviewer for the detailed review and very helpful comments to improve the quality of our manuscript. We highlighted the revised parts in the manuscript file.

1) A flowchart would be helpful to better understand the methodological process followed by the authors.

We thank the Reviewer for the suggestion. The workflow (flow-chart) of the methodological process is provided in Figure 6. We have extended the figure legend for a better understanding of the data processing.

2) The data analyzed are from murine datasets. The authors should comment on the possibility of conducting such studies in humans.

We agree with the Reviewer and we extensively revised the discussion section accordingly and included such a statement as follows:

“The interpretability of the molecular results of this study is further limited to mice, although the methodology can be easily translated to human data. We specifically chose murine RNA-Seq data since the amount of in vivo studies that are available and that provide experimental evidence on gene-to-phenotype associations are greater in mice than in humans.”

3) It is unclear why monocytes were not included in the analysis

We apologize if this was not clear. In the analysis, we included all samples from both datasets, and the first dataset includes monocytes too. Only the second dataset doesn’t include monocyte, but this is because in the study that was chosen (Yoshida et al. Cell, 2019) the only two monocytes subsets didn’t yield any signal in the pile-up traces of ATAC-seq. Furthermore, since an extensive comparison of all results with the literature is not possible, we focused on the most distinct ones from these two datasets

4) Comment on the diagnostic and therapeutic utility of in silico studies and whether they require further in vivo experimental verification is helpful

We thank the Reviewer for this comment and we have extended the discussion section on the utility of in silico studies to predict the key regulators and motifs associated with immune mediators.  

5) It would be interesting to know whether such studies are applicable not only to lipids but to mediators such as cytokines and for whose blockade we have effective tools for treating chronic inflammation

Indeed these studies are also applicable to any mediator other than lipids. Accordingly, in the revised discussion section, we now mention the applicability of our studies to other data and its limitations, as suggested.

6) Several errors regarding missing bibliographic references are pointed out in my review copy

We assume that the reviewer refers to the error messages of figure references in the main text, because we do not have access to the review copy. We apologize for not crosschecking whether the references in the text are conserved during file conversion. We have updated all references and hope that they are now displayed correctly. If the reviewer refers to a different issue, we gladly revise our manuscript again after receiving the comments.

Reviewer 2 Report

The authors present a very interesting manuscript using in silico research to identify regulatory networks of gene expression relative to pro-inflammatory lipid mediators and specialised pro-resolving molecule synthetic pathways. The topic is relevant and novel and fully meets the scope of IJMS. The methodology is interesting and sound, and overall the manuscript is well written and clear. There are however, some issues that should be clarified. 

1. Please correct several automatic text errors that appear throughout the text, probably linking text citations to figures, with the error message "error!reference not found". Please correct this in lines 109, 119, 125, 131, 180, 190, 211, 242, 290, 294, 382, 409, 441, 444, 448, 462, 478. 

2. The acronym UMAP is used before it is described. Please correct accordingly.

3. What was the rationale for the selection of the specific murine single cell RNA sequencing profiles GSE122108 and GSE109125? This should be better justified in the manuscript in the methods and also in the discussion section. 

4. There are several elements and ideas in the results section that should rather be included in the discussion section. For example in line 301: "Egr1 is involved in the response to mechanical or oxidative stress and, thus, the development of atherosclerosis from plaques and hypertonia [60,63,64]. It could be hypothesized that increased expression in aortic macrophages provides a phys iological advantage in responding efficiently to stress stimuli, involving the synthesis of LMs. Overall, the physiological relevance of the difference in the LM gene regulation is not clear." this sentences clearly should be integrated in the discussion. 

5. Discussion should provide a more comprehensive integration with similar literature and also indicate the strengths and limitations of this study. 

Author Response

We thank the Reviewer for the kind appreciation of our manuscript. We highlighted the revised parts in the manuscript file.

1) Please correct several automatic text errors that appear throughout the text, probably linking text citations to figures, with the error message "error!reference not found". Please correct this in lines 109, 119, 125, 131, 180, 190, 211, 242, 290, 294, 382, 409, 441, 444, 448, 462, 478.

We apologize for not crosschecking whether the figure references in the text are conserved during file conversion. We have amended and updated all references and hope that they are now displayed correctly.

2) The acronym UMAP is used before it is described. Please correct accordingly.

We thank the Reviewer for spotting this and we replaced all mentions of “UMAP” before its introduction with “unsupervised machine learning”

3) What was the rationale for the selection of the specific murine single cell RNA sequencing profiles GSE122108 and GSE109125? This should be better justified in the manuscript in the methods and also in the discussion section.

We agree with the Reviewer. To better explain the rationale for the dataset selection, we added additional statements to the method and discussion section explaining the advantages of our study's data in more detail.

4) There are several elements and ideas in the results section that should rather be included in the discussion section. For example in line 301: "Egr1 is involved in the response to mechanical or oxidative stress and, thus, the development of atherosclerosis from plaques and hypertonia [60,63,64]. It could be hypothesized that increased expression in aortic macrophages provides a physiological advantage in responding efficiently to stress stimuli, involving the synthesis of LMs. Overall, the physiological relevance of the difference in the LM gene regulation is not clear." this sentences clearly should be integrated in the discussion.

We thank the Reviewer for this comment. By restricting the comparison with the literature to the results section, we aimed to focus the discussion section on the applicability of the methodology in research. As the presented results argue for our approach, we thought to include their discussion in the result section as well. However, we understand that some parts, especially the one mentioned by the reviewer, are more appropriate for the discussion section. We, therefore, moved some of them to the discussion section.

5) Discussion should provide a more comprehensive integration with similar literature and also indicate the strengths and limitations of this study.

Thank you for the comments. We have now extensively revised the discussion section with the strengths and limitations of our study.